# Hamstring muscle architecture assessed sonographically using wide field of view: A reliability study

**Kevin Cronin** [1]* , **Shane Foley** [1] , **Seán Cournane** [2] , **Giuseppe De Vito** [3] ,
**Eamonn Delahunt** [4,5]

1 School of Medicine, University College Dublin, Dublin, Ireland, 2 School of Physics, University College
Dublin, Dublin, Ireland, 3 Department of Biomedical Sciences, University of Padova, Padua, Italy, 4 School of
Public Health, Physiotherapy and Sports Science, University College Dublin, Dublin, Ireland, 5 Institute for
Sport and Health, University College Dublin, Dublin, Ireland

☯ These authors contributed equally to this work.
* kevin.cronin@ucd.ie

## Abstract

### Purpose

To assess the intra-rater reliability of static wide field of view ultrasound to quantify the architectural characteristics of the hamstring muscles.

### Methods

Twenty amateur male athletes were recruited. Their left hamstring muscles were assessed using static wide field of view ultrasound on two separate occasions. Static ultrasound images were acquired by a single sonographer using a 92mm linear transducer. The architectural characteristics (muscle length, fascicle length, pennation angle and muscle thickness) from two distinct locations of the bicep femoris long head and semimembranosus were evaluated. Muscle length and thickness of the bicep femoris short head and semitendinosus muscle were also evaluated. Intraclass correlation coefficient analyses were performed to determine the intra-rater reliability of the performed measurements.

### Results

Both muscle (intraclass correlation coefficient = 0.99; standard error of measurement = 4.3 to 6.6mm) and fascicle (intraclass correlation coefficient = 0.92 to 0.98; standard error measurement = 1.1 to 2.4mm) length were measured with excellent intra-rater reliability. Muscle thickness was measured with excellent reliability (intraclass correlation coefficient = 0.9 to 0.96; standard error of measurement = 0.91mm to 1.4mm) for all hamstring muscles except for the proximal segments of the bicep femoris short head (intraclass correlation coefficient = 0.85; standard error of measurement = 0.84mm) and semitendinosus (intraclass correlation coefficient = 0.88; standard error of measurement = 0.82mm), which were measured with good reliability. Pennation angle was measured with good reliability (intraclass correlation coefficient = 0.77 to 0.87; standard error of measurement = 1 to 1.6˚).

**Data Availability Statement:** Data cannot be shared publicly because it contains sensitive material (hamstring muscle architecture ultrasound images). Data are available from the University College Dublin Ethics Committee (contact via email

at research.ethics@ucd.ie) for researchers who meet the criteria for access to confidential data.

**Funding:** The author(s) received no specific funding for this work.

**Competing interests:** The authors have declared that no competing interests exist.

## Conclusion

The architectural characteristics of the hamstring muscles of male amateur athletes can be reliably quantified using static wide field of view ultrasound.

## Introduction

Hamstring strains are the most prevalent muscle injuries incurred by athletes participating in field sports with the biceps femoris long head (BFlh) being the most commonly injured hamstring muscle [1, 2]. A self-reported history of previous hamstring strain injury is a primary risk factor for re-injury [3]. The prevalence of hamstring strain re-injury is high among field sport athletes [4], and ranges from 14%–34% within the same competitive season [1, 5].

The geometric distribution of muscle fascicles within a muscle determine its mechanical function and influences its maximal force output and contraction velocity [6]. The angle of trajectory of a muscle fascicle between the superficial aponeurosis and its insertion into the deep aponeurosis is referred to as its pennation angle [7]. Large pennation angles associate with shorter muscle fascicles, which reduces the contractile velocity and excursion range of the muscle [8]. In contrast, small pennation angles associate with longer muscle fascicles, which decreases the physiological cross-sectional area and maximal force output of the muscle [9].

Muscle excursion range is proportional to muscle fascicle length [7]. Indeed, when compared to short muscle fascicles, long muscle fascicles contain more sarcomeres in series [10]. During fast eccentric activity (i.e., the typical mechanism associated with hamstring strain injuries), the muscle tendon unit undergoes active lengthening [9]. During this type of contraction, long muscle fascicles, compared to short muscle fascicles, will exhibit less strain per sarcomere in series [3]. While limited evidence exists to characterise the effect of hamstring muscle architectural characteristics on strain injury, one prospective study reported that a previously strained hamstring muscle possesses shorter muscle fascicles in comparison to an uninjured muscle [3]. Shorter muscle fascicles contain less sarcomeres in series, which will result in a reduced maximal shortening velocity, which could increase the risk of re-injury [3]. Therefore, the ability to accurately quantify the architectural characteristics of the hamstring muscles may assist clinicians in the management of athletes with acute or recurrent hamstring strain injuries.

Brightness mode (B-mode) ultrasound (US) is the most commonly used medical imaging modality to assess the architectural characteristics of skeletal muscle [11]. However, the acquisition of accurate high-quality images of the architectural characteristics of the hamstring muscle group is challenging and operator dependent [12]. Most of the published research using B-mode US to measure the architectural characteristics of the hamstring muscles has used a relatively limited field of view (FOV), which necessitates the use of extrapolation methods to quantify muscle fascicle lengths [11, 13]. These extrapolation methods, which predominantly use linear approximations based upon muscle thickness and pennation angles, do not account for muscle fascicle curvature and are subsequently prone to error [11]. Indeed, most of the published literature focuses on sonographically evaluating the architecture of the BFlh muscle only [11, 14–16], with few studies sonographically assessing the architecture of the bicep femoris short head (BFsh), semitendinosus (ST) and semimembranosus (SM) muscles [17–19].

Extended Field of View (EFOV) techniques have been used recently to permit full fascicle visibility in the BFlh muscle [11]. EFOV techniques create a composite image from a series of sonograms continuously captured as the transducer is dynamically shifted along the muscle. However, the BFlh muscle is a heterogeneous muscle with fascicles orientating themselves in a

non-linear path inserting into a mid-belly aponeurosis [13] and thus, misalignment of the US transducer from the plane of the fascicles may lead to fascicle length errors [20, 21].

Several muscle fascicle tracking algorithms have been developed to quantify the architectural characteristics of skeletal muscle [22–24]; however they are unsuitable to evaluate heterogenous skeletal muscles such as the hamstrings. Recently, a semi-automated tracing algorithm was developed to quantify the architectural characteristics of the hamstrings [25]. This semi-automated tracing software allows the operator to manually trace the architectural characteristics of the hamstrings, permitting precise measurements [25]. This semi-automated tracing software has precisely measured BFlh fascicle lengths (% CV: 0.64 to 1.12), pennation angles (% CV 2.58 to 10.70) and muscle thickness (% CV 0.48 to 2.04). The precision of this semi-automated tracing software demonstrates an improvement on CVs reported in the published literature for measuring fascicle length: CV: 5.9% [26]; CV: 2% [27]; CV: 0–3.8% [28]; CV: 4–7% [29–31]. The precision of this semi-automated tracing software for measuring pennation angle is in line with previous studies using manual linear tracing of muscle architecture (CV, 4–9.8%) [26–31], however more precise than automated tracking software [23]. However, the intra-rater reliability of this semi-automated tracing software has yet to be established. Considering this, the aim of the present study was to assess the intra-rater reliability of this static wide field of view (WFOV) ultrasound technique for quantifying the architectural characteristics of the hamstring muscles. It was hypothesized that excellent repeatability would be obtained for measuring [1] hamstring muscle length; and [2] the architectural characteristics of the hamstring muscles.

## Materials and methods

We designed and conducted a test-retest study. Muscle fascicle length, muscle thickness and pennation angle were assessed at two locations (Zone A and Zone B; Fig 1) in the BFlh and SM muscles. Two locations were chosen as hamstring architecture is different within each zone, where Zone A represents architecture in the proximal portion of the muscle, while Zone B represents architecture in the distal portion of the muscle. Muscle thickness was assessed at two locations (Zone A and Zone B) in the BFsh and ST muscles. The length (i.e., muscle length) of each muscle (BFlh, BFsh, SM, ST) was also assessed (Table 1). Fascicle length and pennation angle in the BFsh and ST muscles was not assessed, as the length of the fascicles extended beyond our WFOV.

Twenty recreationally active men with no history of lower limb muscle injury were recruited via convenience sampling and assessed on two separate occasions (session 1 and session 2; test-retest). For convenience, only males were recruited because they exhibit less subcutaneous and intramuscular adipose tissue in the posterior thigh than females, which allowed for greater sonogram echogenicity and thus hamstring muscle architecture identification [16]. Each session was three weeks apart. Volunteers were requested not to undertake any vigorous lower limb activity during the test-retest period [32, 33]. All participants signed a consent form which was approved by the UCD Human Research Ethics Committee (LS-E-19-83-Cronin-Foley).

All sonograms were captured using a Hitachi Noblus ultrasound scanner (Hitachi Medical Systems, UK) with a 92mm WFOV transducer (Hitachi EUP-L53L). Trapezoidal imaging was implemented at the beginning of each scan to create an image base wider than the footprint of the transducer; this allowed for the inclusion of more lateral muscle architecture, up to 100mm for a depth of 80mm. The primary author (sonographer with >200 hrs hamstring scanning experience) performed all assessments and collated and digitised all sonograms for subsequent analysis. Experienced sonographers, when compared to novice sonographers, generate higher quality sonograms which permits reliable skeletal muscle image acquisition and architectural analysis [34].

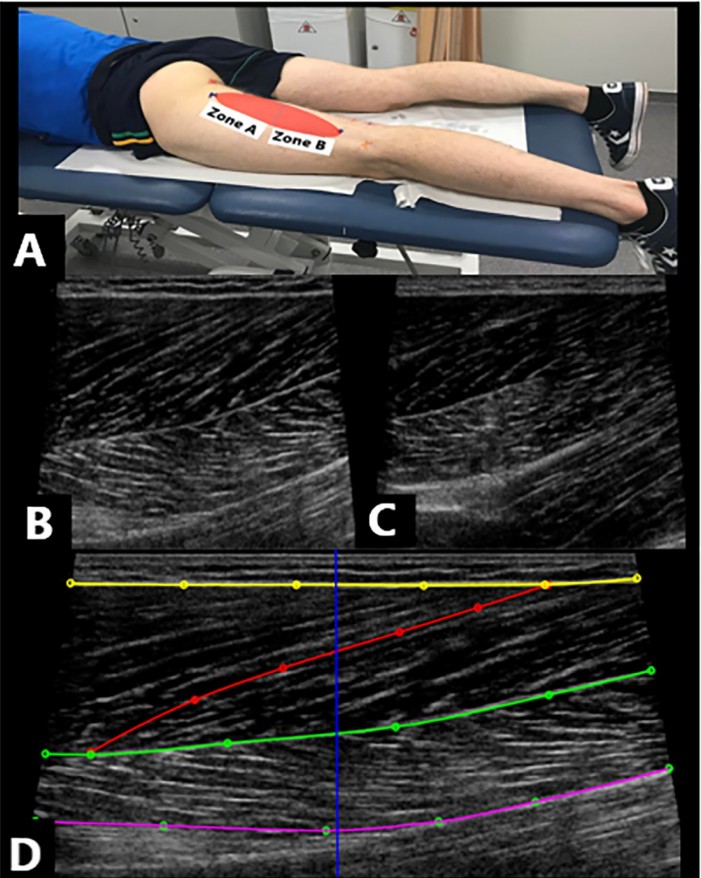

**Fig 1. Procedures used to assess the hamstring muscle architecture sonographically. A:** Identification of the proximal and distal myotendinous junctions to determine a proximal zone (Zone A) and distal zone (Zone B) of the muscle. **B:** a longitudinal section sonogram representing Zone A of the BFlh muscle and **C.** a sonogram representing Zone B of the BFlh muscle. **D**. Quantification of the architectural characteristics of the BFlh muscle via a semi-automated tracking algorithm, (yellow = superficial aponeurosis, red = fascicle length, green = mid muscle aponeurosis, purple = deep aponeurosis, blue = muscle thickness).

## Hamstring US acquisition

During the acquisition of the sonograms, participants were positioned in prone lying on a portable physiotherapy plinth. Participants' legs were adjusted so that the medial aspect of the thigh was aligned with a previously placed mark on the plinth. A small amount of coupling gel

**Table 1. Mean muscle length of the hamstring muscles (session 1 *vs*. session 2).**

| Muscle | Session 1 | Session 2 | ICC (95% CI) | SEM |
|---|---|---|---|---|
| BFlh | 323.5 ± 19.0 | 323.25 ± 19.7 | 0.99 (0.99–0.99) | 4.3 |
| BFsh | 253 ± 24.8 | 251.3 ± 26.5 | 0.99 (0.98–0.99) | 5.7 |
| SM | 317.5 ± 22.7 | 317.5 ± 23.1 | 0.99 (0.99–0.99) | 5.1 |
| ST | 329.5 ± 30.0 | 329.75 ± 30.2 | 0.99 (0.98–0.99) | 6.6 |

Session 1 and session 2 refer to data collected during day 1 and day 2 of the test-retest assessment. Muscle length was measured in mm. Data for session 1 and session 2 are presented as mean ± standard deviation. BFlh = biceps femoris long head. BFsh = biceps femoris short head. SM = semimembranosus. ST = semitendinosus. ICC = intra-class correlation coefficient value. CI = confidence interval. SEM = standard error of measurement.

was placed on the posterior thigh of the left limb of all participants to allow transmission of the US waves intramuscularly. To standardise hamstring ultrasound acquisition, the left limb was only chosen. This was the limb closest to the sonographer, making it technically easier to acquire sonograms.

To identify the proximal and distal myotendinous junctions (MTJs), the technique described by Freitas and colleagues was used [13]. A 'mark' on the skin was used to indicate the position of the MTJs, with the distance between these two marks (in millimetres) being used to quantify muscle length.

## Protocol

Each hamstring muscle was divided into two distinct zones, a proximal zone (Zone A) and distal zone (Zone B). The proximal zone represents architecture adjacent to the proximal MTJ, whereas the distal zone represents architecture adjacent to the distal MTJ. The specific region for the proximal zone was chosen by scanning the muscle in question adjacent to the proximal MTJ in the longitudinal orientation. Once the sonogram clearly illustrated a muscle fascicle inserting into the intermediate/deep aponeurosis, a mark was made on the left-hand side of the transducer. This mark was then measured from the proximal MTJ and recorded so that the transducer could be placed in the same location in the re-test session. The exact same approach was undertaken for the distal zone with the distal MTJ being the feature of interest.

Between each sonogram acquisition the transducer was lifted from the participants' posterior thigh and replaced at the same mark. Three sonograms were acquired at each mark. All sonograms were annotated according to the specific muscle and zone under investigation, anonymised and exported for analysis.

Sonograms were analysed with a semi-automated tracking algorithm to quantify skeletal muscle architecture [25]. This was a custom-developed tracking tool for analysing static ultrasound images developed in Matlab® (MathWorks R2019b, Natick, MA, USA). This semi-automated tracking algorithm permits a precise measurement of the architectural characteristics of the hamstring muscles [25]. It accounts for fascicle curvature, imaging the whole fascicle when using WFOV ultrasound [25]. This contrasts to previously used extrapolation methods to quantify hamstring muscle architecture, which results in an overestimation of fascicle length [11, 35]. The pennation angle is defined as the apex angle of the underlying aponeurosis and the fascicle, taking the fascicle line from a point a distance of 2.5mm along the fascicle [25].

## Quality control

Prior to the study, quality control (QC) tests on the US system were carried out using a tissue equivalent test object (Gammex 403 GS LE precision multipurpose phantom) for assessing uniformity, depth of penetration, contrast resolution, slice thickness, axial, and lateral resolution. Settings on the ultrasound system were standardised to maintain a consistent approach to acquire high quality sonograms. B-mode gain was set at 10 deciBels (dB), Depth was maintained at 80mm, Focus was set central to the region of interest after 40mm. Trapezoidal imaging was utilised to extend the field of view, room lighting was powered off and the portable physiotherapy plinth was set to a height of 0.6m from the floor. All ultrasound scanning was performed while the sonographer was in a seated position.

## Statistical analysis

The sample size was calculated by use of G*Power software. Considering an effect size of 0.9, significance level of 0.05 and a statistical power ($p$) of 0.8, the minimal sample size was 16. In summary, 20 subjects participated in this study, $p = 0.87$. The statistical analysis of hamstring

muscle architecture was performed using SPSS software for Windows, version 26 (IBM, Chicago, USA). Intra-rater reliability values were calculated for: muscle length (BFsh, BFlh, ST, SM); muscle fascicle length (BFlh, SM); muscle thickness (BFsh, BFlh, ST, SM); and pennation angle (BFlh, SM). Intraclass correlation coefficient analyses (ICC $_{3,1}$) (two-way mixed effects, absolute agreement, single rater/measurement) were used to assess the relative homogeneity of hamstring muscle architectural quantification within sessions in relation to the total observer variation between sessions. For this analysis, an absolute agreement was selected. Intraclass correlation coefficients (ICC) were classified as follows: poor reliability (<0.5), moderate reliability (0.5 to 0.75), good reliability (0.75 to 0.9) and excellent reliability (>0.9) [36]. The standard error of measurement (SEM) was determined as the standard deviation of the difference in measurements (SD$_{dif}$) divided by $\sqrt{2}$ using the intra-rater data (n = 20) [37].

## Results

### Test-retest reliability

The respective BFlh, BFsh, SM and ST muscle lengths for each session (i.e., session 1 and session 2) are detailed in Table 1, along with the intra-session repeatability (i.e., intra-rater reliability–ICC value) and the standard error of measurement. The assessment of muscle length for all hamstring muscles (BFlh, BFsh, SM and ST) showed excellent intra-rater reliability.

The intra-session repeatability for the architectural characteristics of BFlh, BFsh, SM and ST are illustrated in Table 2. For muscle fascicle length, all ICCs were almost 1 indicating excellent reliability (Table 2). For muscle thickness all ICC values were above 0.9, indicating excellent reliability, except for the BFsh (good reliability, ICC = 0.85) at Zone A. The ICC

**Table 2. Architectural comparisons of the hamstring muscles (session 1 *vs*. session 2).**

| Parameter | Session 1 | Session 2 | ICC (CL) | SEM |
|---|---|---|---|---|
| Fascicle Length (mm) | | | | |
| BFlh Zone A | 74.3 ± 5.3 | 74.8 ± 4.5 | 0.92 (0.79–0.97) | 1.1 |
| BFlh Zone B | 64.2 ± 10.4 | 64.8 ± 9.8 | 0.98 (0.94–0.99) | 2.3 |
| Semimembranosus Zone A | 56.6 ± 10.8 | 58.2 ± 10.7 | 0.98 (0.92–0.99) | 2.4 |
| Semimembranosus Zone B | 48.5 ± 9.3 | 48.5 ± 8.3 | 0.95 (0.87–0.98) | 2 |
| Muscle Thickness (mm) | | | | |
| BFlh Zone A | 34 ± 4.1 | 34 ± 5.1 | 0.91 (0.76–0.96) | 1 |
| BFlh Zone B | 33.3 ± 4.0 | 33.1 ± 4.2 | 0.93 (0.81–0.97) | 0.91 |
| BFsh Zone A | 19.9 ± 4.7 | 19.9 ± 2.9 | 0.85 (0.61–0.94) | 0.84 |
| BFsh Zone B | 19.4 ± 3.9 | 18.7 ± 3.1 | 0.92 (0.80–0.97) | 0.79 |
| Semimembranosus Zone A | 35.1 ± 6.2 | 35.2 ± 6.5 | 0.96 (0.91–0.99) | 1.4 |
| Semimembranosus Zone B | 35.5 ± 5.2 | 36.7 ± 5.6 | 0.92 (0.80–0.97) | 1.2 |
| Semitendinosus Zone A | 25.8 ± 3.7 | 26.6 ± 3.7 | 0.88 (0.71–0.95) | 0.82 |
| Semitendinosus Zone B | 26.3 ± 4.1 | 27.7 ± 4.1 | 0.90 (0.67–0.97) | 0.92 |
| Pennation Angle (˚) | | | | |
| BFlh Zone A | 24.4 ± 4.6 | 23.4 ± 4.4 | 0.77 (0.41–0.91) | 1 |
| BFlh Zone B | 24.2 ± 5.5 | 24.2 ± 5.9 | 0.87 (0.66–0.95) | 1.3 |
| Semimembranosus Zone A | 26.2 ± 6.4 | 25.9 ± 7.6 | 0.88 (0.67–0.95) | 1.6 |
| Semimembranosus Zone B | 24.9 ± 7.2 | 25.4 ± 7.0 | 0.83 (0.56–0.94) | 1.6 |

Session 1 and Session 2 refer to data collected in day 1 and day 2 of a test-retest assessment. The data for muscle length was acquired and digitised by the same rater. Fascicle length and muscle thickness is measured in millimetres (mm), pennation angle is measured in degrees (˚), data for session 1 and session 2 are presented as mean ± SD. ICC, intraclass correlation coefficient, CL confidence level, SEM, standard error of mean.

values for pennation angle varied between muscle and between zones in the same muscle. Good reliability values were observed for pennation angle at Zone A and Zone B across the entire hamstring muscle group ranging from (0.77–0.88) (Table 2).

## Discussion

We assessed the intra-rater reliability of a static wide field of view (WFOV) ultrasound technique for quantifying the architectural characteristics of the hamstring muscles. Our hypothesis that excellent repeatability would be obtained for measuring hamstring muscle length was fully confirmed. Our hypothesis that excellent repeatability would be obtained for measuring the architectural characteristics of the hamstring muscles was partially confirmed; ICC values for fascicle length ranged from 0.92 to 0.98 (excellent reliability), for muscle thickness ranged from 0.85 to 0.96 (good–excellent reliability), and for pennation angle ranged from 0.77 to 0.88 (good reliability). Our repeatability results for the assessment of BFlh are in agreement with those reported by previous authors [11, 38, 39].

### Muscle length

Our hypothesis that excellent repeatability would be obtained for measuring muscle length of the hamstring muscles was confirmed; ICC 0.98 to 0.99. Our results are similar to some studies measuring hamstring muscle length [17, 40, 41], while some differences are observed when compared to previous studies [19, 42, 43]. Variations in hamstring muscle length is attributed to measurements taken from dissection in cadavers [40–42] and differences in the technique used to quantify muscle length, whereby all previous studies have quantified muscle length by incorporating the muscle-tendon unit [17, 19, 41–43]. In our study we determined muscle length as the distance between the proximal and distal MTJ's, which was similar to the technique described by Freitas and colleagues [13] when assessing BFlh fascicle length (Lf) with ultrasound.

### Muscle thickness

Our hypothesis that excellent repeatability would be obtained for measuring muscle thickness of the hamstring muscles was partially confirmed; ICC values ranged from 0.85 to 0.96 (good–excellent reliability). Previous studies [11, 13, 16, 38, 43] assessing muscle thickness of BFlh have reported excellent reliability. However, to our knowledge, our study is the first to include reliability analyses for the quantification of muscle thickness of BFsh, and SM. Great variation exists around the location of sonographically assessing muscle thickness, which could contribute to differences in muscle thickness across studies [13, 38, 43]. Some authors choose numerous locations within the muscle to assess muscle thickness [17, 43], while other authors measured muscle thickness within the mid-belly of the muscle [38], while some only measured muscle thickness at a specific point where fascicles were clearly visible inserting to the superficial and intermediate aponeurosis [13].

### Fascicle length

We obtained excellent reliability for measuring BFlh and SM fascicle length. This is not an uncommon observation. Chleboun and colleagues [44] reported high inter-rater reliability (ICC = 0.87) for BFlh fascicle length whereas Kellis and colleagues [43] compared cadaveric specimens versus US to provide validation, with high intra-rater reliability values (ICC = 0.9) for BFlh fascicle length and similarly for ST fascicle length (ICC = 0.7 to 0.9). Timmins and colleagues [38] also report high measurement reliability for all architecture characteristics

(ICC = >0.97). Franchi [11] compared static US measurements via two distinct trigonometric extrapolation methods to quantify BFlh muscle architecture versus EFOV yielding excellent reliability for fascicle length (ICC = 0.91 to 0.99).

In our study fascicle length of BFlh ranged from 64.2±10.4mm to 74.8±4.5 mm across zone A and zone B. A substantial amount of heterogeneity exists in the reporting of fascicle length of BFlh in the published literature; values of 57.3mm [45] and 119.4mm [3] have been reported. Similarly, for SM, variation exists in the reporting of fascicle length in the published literature; values of 49mm [42] and 74.9mm [46], with all these measurements sampled on cadaveric specimens [17, 40, 42], have been reported. In our study, fascicle length of SM was 57.4±10.75mm (Zone A) and 48.5±8.8mm (Zone B). For ST, fascicle lengths range from 90–240mm [19, 40–43] while for BFsh fascicle length ranges from 104mm–140mm [17, 40, 42]. We did not measure fascicle length of BFsh or ST, as the length of the fascicles extended beyond our US WFOV.

In our opinion, there are many factors which likely contribute to the heterogeneity of reported fascicle length of the hamstring muscles. Firstly, measurement of fascicle length is dependent on the location and placement of the US transducer. For example, some studies have assessed fascicle length at a position of 50% of a line connecting the ischial tuberosity and knee fold, while other studies have assessed fascicle length on a line connecting the greater trochanter and the tibial condyle [45]. Additionally, some studies have assessed fascicle length at discrete positions along the length of the muscle [43]. Indeed, where the transducer placement is implemented along a muscle, utilising a similar transducer with a FOV (47mm) and the same extrapolation method for measuring fascicle lengths, measurements vary from 81.7mm [39] to 107mm [10]. Our study is unique whereby it assesses the architectural characteristics at two distinct locations using a WFOV technique.

Many previously published studies have used a small FOV when assessing fascicle length of the hamstring muscles [38, 39, 47, 48]. When the FOV is smaller than the actual fascicle length, it is necessary to use WFOV or EFOV. Recently, Pimenta and colleagues [16] identified that BFlh fascicle length was overestimated when using one of the trigonometric extrapolation equations to measure fascicle length. Indeed, similar findings have been echoed by Franchi and colleagues [11] when they compared EFOV US to static-image acquisition, where estimation was larger for longer fascicle lengths on static acquired images. These findings potentially explain why previous data [3, 39] captured on US transducers of 40–50mm FOV's have reported larger fascicle lengths of the hamstring muscles.

Furthermore, some previous studies have reported on fascicle length of the hamstrings from cadaveric sarcopenic tissue [43] making direct comparisons to *in vivo* studies difficult. The heterogeneity in fascicle length of the hamstring muscles reported in the published literature illustrated how differences in the site and mode of data acquisition influence the reported outcomes. As such, our transparent and replicable standardisation of the acquisition site may facilitate more accurate comparisons across studies in the future, as well as the longitudinal follow-up of athletes to assess responses to exercise and/or injury.

## Pennation angle

Our hypothesis that excellent repeatability would be obtained for measuring pennation angle of the hamstring muscles (BFlh and SM) was somewhat confirmed; ICC: 0.77 to 0.88 (good reliability). Substantial variation exists in the reporting of hamstring muscle pennation angle in the published literature. For the SM, values ranging from 10° to 31° [40, 49] have been reported. For BFsh, values ranging from 13° to 25° [17, 40] have been reported. For the BFlh, values ranging from 7° to 28° [46, 49] have been reported. The fascicle orientation in the ST

muscle is parallel yet values reported range from 0˚ to 13˚ [43, 49]. It is likely that disparities in US technique, image analysis, and *in vivo* imaging in contrast to cadaveric specimens are accountable for this substantial heterogeneity.

Of particular note, all previously published data on hamstring muscle pennation angle has failed to detail the specific distance along the fascicle at which the pennation angle was evaluated [3, 11, 13, 16, 18, 39, 44, 45, 47, 48, 50]. In our study, we standardized the assessment of pennation angle to 2.5mm along the fascicle inserting into the intermediate aponeurosis for the BFlh muscle and the SM muscle [25]. We choose 2.5mm along the fascicle, as beyond this point the pennation angle increases considerably, and does not represent the angle of the fascicle inserting into the aponeurosis. In this study we yielded good repeatability with PA for BFlh and SM muscles (Table 2). This is similar to PA reliability values in previous studies [11, 13, 16]. We did not measure the pennation angle of BFsh or ST, as the length of the fascicles extended beyond our WFOV.

[17,19, 40–43] EFOV is an ultrasound technique commonly used to image the BFlh muscle [11, 15]. EFOV requires a high operator skill that stitches a series of ultrasound images together to create a composite image [35]. A recent study conducted by Franchi and colleagues (2020), reported improved intra-session reliability for EFOV imaging of the BFlh muscle and further, demonstrated that extrapolation methods performed using limited US FOV coverage led to overestimation of Lf measurements, as compared with EFOV [35]. However, the hamstring muscle architecture is extremely heterogeneous [12], and any misalignment of the US transducer from the plane of the curved fascicles would most likely lead to fascicle length errors [20, 21]. Indeed, not all US application systems support EFOV. The ability to trend hamstring architectural characteristics is dependent on the accuracy and repeatability of a measurement technique. A potential approach for overcoming the aforementioned US imaging limitations is through employing a transducer with a WFOV that captures the entire fascicle length, avoiding the need for extrapolation methods. Our results indicate that WFOV is a reliable method to measure the fascicle lengths of the BFlh and SM muscles. WFOV is unable to measure the fascicle lengths of the BFsh and ST muscles as their fascicles extend beyond the US transducers FOV.

The following limitations should be considered. WFOV US is a 2D technique, and because of this we cannot account for 3D curvature [12]. However by utilising a semi-automated tracing software tool, we were able to account for fascicle curvature, and precisely quantify the architectural characteristics of the hamstring muscles. Minimal pressure was applied to the ultrasound transducer when acquiring sonograms. However, transducer pressure was not quantified during the study, and it is important to acknowledge that pennation angle may be under-estimated when too much pressure is applied to the US transducer [51]. Finally, it is important to acknowledge that FOV is a technical limitation that requires extrapolation methods to be used.

## Conclusion

Our results indicate that the architectural characteristics of the hamstring muscles of male amateur athletes can be reliably quantified using static wide field of view ultrasound.

## Author Contributions

**Conceptualization:** Kevin Cronin, Shane Foley, Eamonn Delahunt.

**Data curation:** Kevin Cronin, Shane Foley.

**Formal analysis:** Kevin Cronin, Shane Foley, Eamonn Delahunt.

**Funding acquisition:** Kevin Cronin.

**Investigation:** Kevin Cronin.

**Methodology:** Kevin Cronin, Seán Cournane, Giuseppe De Vito, Eamonn Delahunt.

**Project administration:** Kevin Cronin.

**Resources:** Kevin Cronin.

**Software:** Kevin Cronin, Seán Cournane.

**Supervision:** Kevin Cronin, Shane Foley, Seán Cournane, Giuseppe De Vito, Eamonn Delahunt.

**Validation:** Kevin Cronin, Eamonn Delahunt.

**Visualization:** Kevin Cronin, Eamonn Delahunt.

**Writing – original draft:** Kevin Cronin.

**Writing – review & editing:** Kevin Cronin, Shane Foley, Seán Cournane, Giuseppe De Vito, Eamonn Delahunt.

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
