## [Decision Letter · Decision Letter 0]

26 Aug 2022

PONE-D-22-20555Hamstring muscle architecture assessed sonographically using wide field of view: a reliability studyPLOS ONE

Dear Dr. Cronin,

Thank you for submitting your manuscript to PLOS ONE. After careful consideration, we feel that it has merit but does not fully meet PLOS ONE’s publication criteria as it currently stands. Therefore, we invite you to submit a revised version of the manuscript that addresses the points raised during the review process.

 Both reviewers found merit in your manuscript, however, each provided suggestions that can improve the manuscript. There are places throughout the manuscript that reviewers felt could be made clearer.

We look forward to receiving your revised manuscript.

Kind regards,

Jeremy P Loenneke

Academic Editor

PLOS ONE

Journal Requirements:

Reviewers' comments:

Reviewer's Responses to Questions

**Comments to the Author**

1. Is the manuscript technically sound, and do the data support the conclusions?

Reviewer #1: Partly

Reviewer #2: Yes

2. Has the statistical analysis been performed appropriately and rigorously? 

Reviewer #1: I Don't Know

Reviewer #2: Yes

3. Have the authors made all data underlying the findings in their manuscript fully available?

Reviewer #1: Yes

Reviewer #2: No

4. Is the manuscript presented in an intelligible fashion and written in standard English?

Reviewer #1: Yes

Reviewer #2: Yes

5. Review Comments to the Author

Reviewer #1: COMMENTS FOR THE AUTHOR(S)

Thank you for the opportunity to review the paper “Hamstring muscle architecture assessed sonographically using wide field of view: a reliability study”. The purpose of this study was to assess the reliability of static wide field of view ultrasound to quantify the muscle architecture of the hamstring muscles: Bicep Femoris long head, Bicep Femoris short head, Semitendinosus and Semimembranosus. The authors recruited 20 male amateur athletes and performed ultrasound measurements on two separate occasions. They concluded that static wide field of view is a reliable ultrasound technique to quantify the architectural characteristics of the hamstring muscles. Although the manuscript is very interesting and well written, some concerns about the sample size need to be point out and additional information are necessary.

General comment

• Why only 10 participants were recruited? Do the authors have any information to support that this sample size has sufficient statistical power for this reliability study? I believe this information is very relevant, and it is the main point to increase (or decrease) the support for the conclusions reported by the authors.

Specific comments

Abstract

Lines 51 and 52: I suggest being specific for the population investigated in the present study (male amateur athletes).

Introduction

•No comment.

Materials and Methods

•Line 141 – Why only 20 participants? Do the authors have any information to support that this sample size has sufficient statistical power for this reliability study?

•Line 141 – Why only men? Is there some rationale for not include women?

Statistical Analysis

•Line 212 – Is “3,1” a typo? Or it is supposed to be the reference number 31?

Discussion

•There is a mix of references styles over this section. For example, “(18, 19, 22, 33, 34)”, “(13) (17)”, “Chleboun et al. (2001)”, “Timmins and colleagues (2015)”. I suggest using only one reference style accordingly PlosONE authors guidelines.

Conclusion

•I suggest being specific for the population investigated in the present study (male amateur athletes).

Reviewer #2: General Comments

The authors stated the purpose of this article was to assess the reliability of wide field of view ultrasound in architecture of the four hamstring muscles. In general, the paper is methodologically sound however, improvements on clarity and some section-specific adjustments are warranted. Yet, the manuscript could be both useful for researchers and clinicians with the following adjustments taken into consideration.

Specific Comments

Introduction

1. The first paragraph is choppy (Lines 72-76), could use better transition words between sentences

Example: (Line 73) “…sports; the biceps femoris long head (BFlh) is the most…”

May sound better as: “…sports with the biceps femoris long head (BFlh) being the most…”

2. Lines 79-81: Citation warrented.

3. Lines 83-84: All pennation angles are acute by design and therefore “acute pennation angles” should be changed to a relative term. Further, the sentence starting with “In contrast...” is incorrect. Unsure if this is a typing error or a misunderstanding. Larger pennation angles = greater physiological cross-sectional area (PSCA) and maximal force output. It currently reads as smaller angles have greater PCSA and maximal force output.

4. Lines 100-102: Remove “to image” mid-sentence.

5. Line 122: Elaborate why semi-automated tracing is good (e.g. level of control, etc). The reader must imply when it should be explicitly stated by authors.

6. Line 124: insert change “tracing software precisely measured BFlh” to “tracing software has precisely measured BFlh” – also %CV values are provided here. Please cite or state whether these are within lab values.

Methods

1. Figure 1: Please label Zones A and B on panel A (not just in legend).

2. Line 143: Justify three-week session gap.

3. Lines 150-151: Provide a citation to support why sonography experience is important. (e.g., Carr et al. 2021 J Funct Morphol Kinesiol)

Hamstring US Acquisition

4. The imaged limb is never mentioned outside of the abstract. State that the left limb was imaged for all participants and justify. Unclear if the left limb was non-dominant for everyone.

Protocol

1. Figure 1: Does the blue line include the skin? Color intensity should be higher and lines could be thicker. Additionally, the image quality of the entire figure is poor.

Quality Control

1. Line 204: Clarify the where the focus was set. After 40mm deep, include (depth 80mm) and place comma after. Unclear is trapezoidal should be next listed item without comma.

2. Line 206: Clarify that the sonographer was in the seated position not the participant.

Statistical analysis

1. Lines 210-211: Would be helpful to see paratheses of muscles after each variable.

Example: “…calculated for: muscle length (BFlh, SM, BFsh, ST), muscle fascicle length (BFlh, SM)…”

Results

1. Line 226: Authors mention “intra-session reliability” in Table 1. Where are the intra-session reliability results? All tables are inter-session… Please clarify. Include both if you have intra- and inter-session.

Discussion

1. Overall discussion could be clearer/better structured. Perhaps subheadings for each variable (thickness, fascicle length, etc.) would be helpful for reader.

2. Line 282: Clarify where the “great variation” around location exists. Is it the individual’s muscle or sonographer? May warrant citation.

3. Lines 301-302 and 346-347: These statements should be in the methods.

4. Are there previous studies that have used the same sites as your study? Unclear. However, the lack of standardization across the literature is certainly an area for growth/improvement as authors have stated.

5. Lines 309-312: Clarify this sentence. The fascicle is longer than the FOV, 81.7-107mm vs 47mm. Was an extended FOW used?

6. Lines 315-319: The pitfalls of trigonometric extrapolation should be stated in methods.

7. Line 319: “..greater..”. Do the authors mean “better” or “larger”? The authors may be trying to explain that static images are overestimating fascicle lengths because they use extrapolation compared to EFOV which depicts entire fascicle and does not need extrapolation…but later (line 363) mentions underestimation. Please rework sentence. Are “static images” regular FOV?

8. Line 321: replace “large” with “larger”

9. Line 322: insert comma after “Furthermore”

10. Line 342: State the pennation angle measurement site in the methods.

11. Line 356: “Lf”, first abbreviation state full term.

12. Line 366: Do authors mean “trace” instead of “trend”?

13. Line 375: Citation uses brackets instead of paratheses.

14. Important to note that FOV is an ultrasound/probe limitation. Not everyone uses the same equipment and therefore there is a variety of ways in which researchers measure architecture.

6. PLOS authors have the option to publish the peer review history of their article (what does this mean?). If published, this will include your full peer review and any attached files.

Reviewer #1: No

Reviewer #2: No

---

## [Author Response · Author response to Decision Letter 0]

16 Sep 2022

UCD School of Medicine 

UCD Health Sciences Centre

Belfield, Dublin 4, Ireland

T +353 1 716 6541

T +353 872036769

Scoil an Leighis agus Eolaíocht 

Ionad Eolaíocht Sláinte UCD

Belfield, Báile Átha Cliath 4, Éire

Kevin.cronin@ucd.ie

www.ucd.ie/medicine

07/09/2022

Re: PONE-D-22-20555

Dear Editors and Reviewers,

Please find enclosed a revised version of our manuscript entitled “Hamstring muscle architecture assessed sonographically using wide field of view: a reliability study” by authors Mr Kevin Cronin, Professor Eamonn Delahunt, Dr Shane Foley, Professor Giuseppe de Vito and Dr Sean Cournane to be considered for publication in PLOS ONE.

We would like to acknowledge Jeremy P Loenneke and anonymous reviewers #1 and #2 for their comments and suggestions which we feel added to and improved our manuscript. We considered the comments with care and acted on the majority of the points raised by the reviewers. Below, please find our answers to all reviewer comments, presented in the same order as in the review.

Editorial Office Comments

Comment 

Author 

Response

The manuscript now meets PLOS ONE’s style

Comment

We note that you have stated that you will provide repository information for your data at acceptance. Should your manuscript be accepted for publication, we will hold it until you provide the relevant accession numbers or DOIs necessary to access your data. If you wish to make changes to your Data Availability statement, please describe these changes in your cover letter and we will update your Data Availability statement to reflect the information you provide

Author 

Response

We will not provide access to repository information (data) as it contains patient identifiers.

We have updated our cover letter.

Comment

If applicable, we recommend that you deposit your laboratory protocols in protocols.io to enhance the reproducibility of your results. Protocols.io assigns your protocol its own identifier (DOI) so that it can be cited independently in the future

Authors

Response

We have uploaded our protocol. 

DOI: dx.doi.org/10.17504/protocols.io.5qpvorxjdv4o/v1 (Private link for reviewers: https://www.protocols.io/private/580657102EBC11ED8A020A58A9FEAC02

Reviewer 1 Comments 

Comment

Why only 10 participants were recruited? Do the authors have any information to support that this sample size has sufficient statistical power for this reliability study? I believe this information is very relevant, and it is the main point to increase (or decrease) the support for the conclusions reported by the authors.

Author 

Response

We recruited 20 participants. We supported this with additional information in the “Manuscript” document:

Line 250 - 252: 

The sample size was calculated by use of G*Power software. Considering an effect size of 0.9, significance level of 0.05 and a statistical power (p) of 0.8, the minimal sample size was 16. In summary, 20 subjects participated in this study, p = 0.87.

Comment

Abstract Lines 51 and 52: I suggest being specific for the population investigated in the present study (male amateur athletes).

Author

Response

We agree. The following has been added:

Line 59 – 60:

The architectural characteristics of the hamstring muscles of male amateur athletes can be reliably quantified using static wide field of view ultrasound.

Comment

Materials and Methods 

Line 141 – Why only 20 participants? Do the authors have any information to support that this sample size has sufficient statistical power for this reliability study?

Author Response

We agree, we have added information on our sample size. The following has been added:

Line 250 - 252: 

The sample size was calculated by use of G*Power software. Considering an effect size of 0.9, significance level of 0.05 and a statistical power (p) of 0.8, the minimal sample size was 16. In summary, 20 subjects participated in this study, p = 0.87.

Comment

Line 141

 – Why only men? Is there some rationale for not include women?

Authors Response

We agree, we have added information why we only included male athletes/rationale for omitting female athletes. The following has been added:

Line 167 – 170:

For convenience, only males were recruited because they exhibit less subcutaneous and intramuscular adipose tissue in the posterior thigh than females, which allowed for greater sonogram echogenicity and thus hamstring muscle architecture identification (16).

Comment

Statistical Analysis

Line 212 – Is “3,1” a typo? Or it is supposed to be the reference number 31?

Authors Response

No, this is the specific type of Intraclass correlation coefficient that is used.

Two-way mixed effects, consistency, single rater/measurement 

https://www.ncbi.nlm.nih.gov/pmc/articles/PMC4913118/pdf/main.pdf

Comment

Discussion

There is a mix of reference styles over this section. For example, “(18, 19, 22, 33, 34)”, “(13) (17)”, “Chleboun et al. (2001)”, “Timmins and colleagues (2015)”. I suggest using only one reference style accordingly PlosONE authors guidelines.

Authors

Response

We have fixed any referencing issues. 

Comment

Conclusion

I suggest being specific for the population investigated in the present study (male amateur athletes).

Authors Response

We agree. The following has been added:

Line 438 – 439:

Our results indicate that the architectural characteristics of the hamstring muscles of male amateur athletes can be reliably quantified using static wide field of view ultrasound. 

Reviewer 2 Comments

Comment

Introduction

1. The first paragraph is choppy (Lines 72-76), could use better transition words between sentences Example: (Line 73) “…sports; the biceps femoris long head (BFlh) is the most…” May sound better as: “…sports with the biceps femoris long head (BFlh) being the most…”

Author Response

We agree. We have rephrased the opening paragraph.

Lines 81 – 86:

Hamstring strains are the most prevalent muscle injuries incurred by athletes participating in field sports, with the biceps femoris long head (BFlh) being the most commonly injured hamstring muscle (1,2). A self-reported history of previous hamstring strain injury is a primary risk factor for re-injury (3). The prevalence of hamstring strain re-injury is high among field sport athletes (4), and ranges from 14% - 34% within the same competitive season (1,5).

Comment

Lines 79-81: Citation warranted.

Author Response

We agree. We have added a citation. It now reads:

Lines 88 – 90:

The angle of trajectory of a muscle fascicle between the superficial aponeurosis and its insertion into the deep aponeurosis is referred to as its pennation angle (7)

Comment

Lines 83-84: All pennation angles are acute by design and therefore “acute pennation angles” should be changed to a relative term.

Further, the sentence starting with “In contrast...” is incorrect. Unsure if this is a typing error or a misunderstanding. Larger pennation angles = greater physiological cross-sectional area (PSCA) and maximal force output. It currently reads as smaller angles have greater PCSA and maximal force output.

Author Response

We agree. This was a typing error. It now reads:

Lines 90 – 94:

Large pennation angles associate with shorter muscle fascicles, which reduces the contractile velocity and excursion range of the muscle (8). In contrast, small pennation angles associate with longer muscle fascicles, which decreases the physiological cross-sectional area and maximal force output of the muscle (9).

Comment

4. Lines 100-102: Remove “to image” mid-sentence.

Author Response

We agree. We have now removed “to image”. It now reads: 

Lines 111 – 113:

However, the acquisition of accurate high-quality images of the architectural characteristics of the hamstring muscle group is challenging and operator dependent (12).

Comment

5. Line 122: Elaborate why semi-automated tracing is good (e.g. level of control, etc). The reader must imply when it should be explicitly stated by authors.

Author Response

We agree. We have now added in additional information. It now reads:

Lines: 134 – 139:

Recently, a semi-automated tracing algorithm was developed to quantify the architectural characteristics of the hamstrings (25). This semi-automated tracing software allows the operator to manually trace the architectural characteristics of the hamstrings, permitting precise measurements (25). This semi-automated tracing software has precisely measured BFlh fascicle lengths (% CV: 0.64 - 1.12), pennation angles (% CV 2.58 - 10.70) and muscle thickness (% CV 0.48 - 2.04).

Comment

6. Line 124: insert change “tracing software precisely measured BFlh” to “tracing software has precisely measured BFlh” – also %CV values are provided here. Please cite or state whether these are within lab values.

Authors Response

We agree. It now reads:

Line 137 – 145:

This semi-automated tracing software has precisely measured BFlh fascicle lengths (% CV: 0.64 - 1.12), pennation angles (% CV 2.58 - 10.70) and muscle thickness (% CV 0.48 - 2.04). The precision of this semi-automated tracing software demonstrates an improvement on CVs reported in the published literature for measuring fascicle length: CV: 5.9% (26); CV: 2% (27); CV: 0 – 3.8% (28); CV: 4 – 7% (29-31). The precision of this semi-automated tracing software for measuring pennation angle is in line with previous studies using manual linear tracing of muscle architecture (CV, 4 – 9.8%) (26-31) however more precise than automated tracking software’s (23)

Comment

Methods 

1. Figure 1: Please label Zones A and B on panel A (not just in legend).

Authors Response

We agree. We have made changes to labels in Figure 1. We have improved the quality of Figure 1.

Comment

2. Line 143: Justify three-week session gap.

Authors Response

We agree. We have added further information and a citation here. It now reads:

Lines: 170 – 171:

Each session was three weeks apart. Volunteers were requested not to undertake any vigorous lower limb activity during the test-retest period (32,33).

Comment

3. Lines 150-151: Provide a citation to support why sonography experience is important. (e.g., Carr et al. 2021 J Funct Morphol Kinesiol)

Authors Response

We agree. We have added a citation and additional information. It now reads:

Lines 179 – 183:

The primary author (sonographer with >200 hrs hamstring scanning experience) performed all assessments and collated and digitised all sonograms for subsequent analysis. Experienced sonographers, when compared to novice sonographers, generate higher quality sonograms which permits reliable skeletal muscle image acquisition and architectural analysis.

Comment

Hamstring US Acquisition 4. The imaged limb is never mentioned outside of the abstract. State that the left limb was imaged for all participants and justify. Unclear if the left limb was non-dominant for everyone.

Authors 

Response

We agree. We have added additional information. It now reads:

Lines 188 – 192:

A small amount of coupling gel was placed on the posterior thigh of the left limb of all participants to allow transmission of the US waves intramuscularly. To standardise hamstring ultrasound acquisition, the left limb was only chosen. This was the limb closest to the sonographer, making it technically easier to acquire sonograms.

Comment

Protocol 1. Figure 1: Does the blue line include the skin? Color intensity should be higher, and lines could be thicker. Additionally, the image quality of the entire figure is poor.

Authors

Response

The blue line does not include the skin. This is well documented in the article referenced.

We have improved the image quality. 

Comment

Quality Control 1. Line 204: Clarify the where the focus was set. After 40mm deep, include (depth 80mm) and place comma after. Unclear is trapezoidal should be next listed item without comma.

Authors Response

We agree. We have added additional information and made the recommended changes. It now reads:

Lines: 244 – 247:

Focus was set central to the region of interest after 40mm. Trapezoidal imaging was utilised to extend the field of view, room lighting was powered off and the portable physiotherapy plinth was set to a height of 0.6m from the floor. All ultrasound scanning was performed while the sonographer was in a seated position. 

Comment

2. Line 206: Clarify that the sonographer was in the seated position not the participant.

Authors

Response

We agree. We have added additional information. It now reads:

Lines: 246 – 247:

All ultrasound scanning was performed while the sonographer was in a seated position. 

Comment

Statistical analysis

1. Lines 210-211: Would be helpful to see paratheses of muscles after each variable. Example: “…calculated for: muscle length (BFlh, SM, BFsh, ST), muscle fascicle length (BFlh, SM)…”

Authors Response

We agree. We have added additional information. It now reads:

Lines: 254 – 256:

Intra-rater reliability values were calculated for: muscle length (BFsh, BFlh, ST, SM); muscle fascicle length (BFlh, SM); muscle thickness (BFsh, BFlh, ST, SM); and pennation angle (BFlh, SM).

C

Results

1. Line 226: Authors mention “intra-session reliability” in Table 1. Where are the intra-session reliability results? All tables are inter-session… Please clarify. Include both if you have intra- and inter-session.

A

These are all intra-rater reliability values. There was only one rater, meaning intrarater reliability was assessed.

C

Discussion 1. Overall discussion could be clearer/better structured. Perhaps subheadings for each variable (thickness, fascicle length, etc.) would be helpful for reader.

A

 We have made the necessary changes to structure the discussion.

Lines: 309 – 435

C

2. Line 282: Clarify where the “great variation” around location exists. Is it the individual’s muscle or sonographer? May warrant citation.

A

We agree. We have added additional information and the appropriate citations. It now reads:

Lines: 328 – 334:

Great variation exists around the location of sonographically assessing muscle thickness, which could contribute to differences in muscle thickness across studies (13,38,43). Some authors have chosen various locations within the muscle to assess muscle thickness, while other authors have measured muscle thickness within the mid-belly of the muscle (38), while some have only measured muscle thickness at a specific point where fascicles were clearly visible inserting to the superficial and intermediate aponeurosis (13).

C

3. Lines 301-302 and 346-347: These statements should be in the methods.

301-302: We did not measure fascicle length of BFsh or ST, as the length of 302 the fascicles extended beyond our US wide field of view.

346-347: We did not measure pennation angle of BFsh or ST, as the length of the 347 fascicles extended beyond our wide field of view.

A

We agree. We have added additional information to the methods. It now reads:

Lines: 161 -163

Fascicle length and pennation angle in the BFsh and ST muscles was not assessed, as the length of the fascicles extended beyond our wide field of view.

C

Are there previous studies that have used the same sites as your study? Unclear. However, the lack of standardization across the literature is certainly an area for growth/improvement as authors have stated.

A

We agree. We have elaborated on your comment and added an additional sentence. It now reads:

Lines: 367 -368

Our study is unique as it assesses the architectural characteristics at two distinct locations using a WFOV technique.

C

5. Lines 309-312: Clarify this sentence. The fascicle is longer than the FOV, 81.7-107mm vs 47mm. Was an extended FOW used?

A

We agree. We have elaborated on your comment and added clarity to this statement. It now reads:

Lines: 364 -367

Indeed, where the transducer placement is implemented along a muscle, utilising a similar transducer with a FOV (47mm) and the same extrapolation method for measuring fascicle lengths, measurements vary from 81.7mm (39) to 107mm (10).

C

6. Lines 315-319: The pitfalls of trigonometric extrapolation should be stated in methods.

A

We agree. We have added additional information and the appropriate citations to the methods section. It now reads:

Lines: 219 – 221:

This is in contrast to previously used extrapolation methods to quantify hamstring muscle architecture, which results in an overestimation of fascicle length (11, 35).

C

7. Line 319: “..greater..”. Do the authors mean “better” or “larger”? The authors may be trying to explain that static images are overestimating fascicle lengths because they use extrapolation compared to EFOV which depicts entire fascicle and does not need extrapolation…but later (line 363) mentions underestimation. Please rework sentence. Are “static images” regular FOV?

A

We agree. We have reworked this statement. It now reads:

Lines: 412 - 415

A recent study conducted by Franchi and colleagues (2020), reported increased intra-session reliability for EFOV imaging of the BFlh muscle and further, demonstrated that extrapolation methods performed using limited US FOV coverage led to overestimation of Lf measurements, as compared with EFOV.

C

8. Line 321: replace “large” with “larger”

A

We agree. It now reads:

Lines: 376 - 378

“captured on US transducers of 40 - 50mm FOV’s have reported larger fascicle lengths of the hamstring muscles…”

C

9. Line 322: insert comma after “Furthermore”

A

We agree. It now reads:

Line: 380

“Furthermore, some previous studies have reported on fascicle length...”

C

10. Line 342: State the pennation angle measurement site in the methods.

A

We agree. We have added additional information and the appropriate citations to the methods section. It now reads:

Lines: 221 – 223:

The pennation angle is defined as the apex angle of the underlying aponeurosis and the fascicle, taking the fascicle line from a point a distance 2.5mm along the fascicle (25).

C

11. Line 356: “Lf”, first abbreviation state full term.

A

We agree. It now reads:

Lines: 31 – 319:

 “when assessing BFlh fascicle length (Lf) with ultrasound....”

C

12. Line 366: Do authors mean “trace” instead of “trend”?

A

No, we mean trend. We wish to trend architecture lengths overtime, thus we require an accurate and repeatable measurement technique. It now reads:

Lines: 418 – 420:

The ability to trend hamstring architectural characteristics is dependent on the accuracy and repeatability of a measurement technique.

C

13. Line 375: Citation uses brackets instead of paratheses.

A

We agree. Citation corrected. It now reads:

Lines: 427 – 428:

The following limitations should be considered. WFOV US is a 2D technique, and because of this we cannot account for 3D curvature (12)

C

14. Important to note that FOV is an ultrasound/probe limitation. Not everyone uses the same equipment and therefore there is a variety of ways in which researchers measure architecture.

A

We agree. We have added an additional statement in the final paragraph. It now reads:

Lines: 433 – 435:

Finally, it is important to acknowledge that FOV is a technical limitation that requires extrapolation methods to be used.

We hope that the revisions in the manuscript will be sufficient to make our manuscript suitable for publication in PLOS ONE.

Yours sincerely

Assistant Professor/Lecturer, Diagnostic Imaging, 

School of Medicine,

University College Dublin

---

## [Decision Letter · Decision Letter 1]

27 Oct 2022

Hamstring muscle architecture assessed sonographically using wide field of view: a reliability study

PONE-D-22-20555R1

Dear Dr. Cronin,

We’re pleased to inform you that your manuscript has been judged scientifically suitable for publication and will be formally accepted for publication once it meets all outstanding technical requirements.

Kind regards,

Jeremy P Loenneke

Academic Editor

PLOS ONE

Additional Editor Comments (optional):

Reviewers' comments:

Reviewer's Responses to Questions

**Comments to the Author**

1. If the authors have adequately addressed your comments raised in a previous round of review and you feel that this manuscript is now acceptable for publication, you may indicate that here to bypass the “Comments to the Author” section, enter your conflict of interest statement in the “Confidential to Editor” section, and submit your "Accept" recommendation.

Reviewer #1: All comments have been addressed

2. Is the manuscript technically sound, and do the data support the conclusions?

Reviewer #1: Yes

3. Has the statistical analysis been performed appropriately and rigorously? 

Reviewer #1: Yes

4. Have the authors made all data underlying the findings in their manuscript fully available?

Reviewer #1: Yes

5. Is the manuscript presented in an intelligible fashion and written in standard English?

Reviewer #1: Yes

6. Review Comments to the Author

Reviewer #1: The authors have adequately addressed all comments raised in a previous round of review. I have no more comments.

7. PLOS authors have the option to publish the peer review history of their article (what does this mean?). If published, this will include your full peer review and any attached files.

Reviewer #1: No
